# How does Weight Correlation Affect the Generalisation Ability of Deep Neural Networks?

**Gaojie Jin**[1], **Xinping Yi**[1], **Liang Zhang**[2,3], **Lijun Zhang**[2,4], **Sven Schewe**[1], **Xiaowei Huang**[1]

[1] University of Liverpool, Liverpool, UK
[2] State Key Laboratory of Computer Science, Institute of Software, CAS, Beijing, China
[3] University of Chinese Academy of Sciences, Beijing, China
[4] Institute of Intellegence Software, Guangzhou, China
{g.jin3, xinping.yi, svens, xiaowei.huang}@liverpool.ac.uk
{zhangliang, zhanglj}@ios.ac.cn

## Abstract

This paper studies the novel concept of *weight correlation* in deep neural networks and discusses its impact on the networks' generalisation ability. For fully-connected layers, the weight correlation is defined as the average cosine similarity between weight vectors of neurons, and for convolutional layers, the weight correlation is defined as the cosine similarity between filter matrices. Theoretically, we show that, weight correlation can, and should, be incorporated into the PAC Bayesian framework for the generalisation of neural networks, and the resulting generalisation bound is monotonic with respect to the weight correlation. We formulate a new complexity measure, which lifts the PAC Bayes measure with weight correlation, and experimentally confirm that it is able to rank the generalisation errors of a set of networks more precisely than existing measures. More importantly, we develop a new regulariser for training, and provide extensive experiments that show that the generalisation error can be greatly reduced with our novel approach.

## 1 Introduction

Evidence in neuroscience has suggested that correlation between neurons plays a key role in the encoding and computation of information in the brain (Cohen and Kohn, 2011; Kohn and Smith, 2005). In deep neural networks (DNNs), or networks for simplicity, the correlation between neurons can be materialised as the correlation between weight matrices of either neurons (for fully-connected layers) or their filters (for convolutional layers), where it is referred to as weight correlation (WC).

WC is, while intriguing, not a concept that is prima facie a primary benchmark for networks. We will, however, provide evidence that it correlates with the generalisation ability of networks—one of the most important concepts in machine learning that reflects how accurately a learning algorithm is able to predict over previously unseen data. The key concept of generalisation ability can be quantified by the generalisation error (GE). Many factors have been discussed (Zhang et al., 2017) related to the GE, including the Lipschitz constant, the smoothness of the loss function, and memorisation, to name but a few. To the best of our knowledge, however, no research that explicitly considers how, and to what extend, a correlation concept—either the WC or any other correlation—affects the GE has been conducted yet. Our observation that the WC correlates positively with GE thus opens an exciting new avenue to reduce the GE, and thus to improve the generalisation ability of networks.

Our hypothesis of positive correlation between WC and GE is motivated by an observation made from Figure 1, which shows such a correlation between WC and GE. Broadly speaking, the GE increases with the WCs.

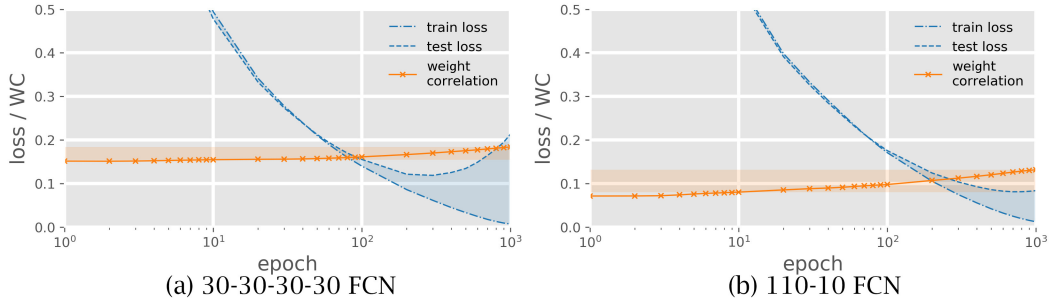

(a) 30-30-30-30 FCN        (b) 110-10 FCN

Figure 1: **(a)** A fully-connected network (of structure 30-30-30-30), which has a large GE. **(b)** A fully-connected network (of structure 110-10), which has a small GE. Note that the curves are presented with log scale. For better presentation, we split the training process into three phases, according to the training epochs: $10^0 \sim 10^1$, $10^1 \sim 10^2$, and $10^2 \sim 10^3$. For **(a)**, the average WCs are 0.152, 0.157, and 0.175, and for **(b)**, the average WCs are 0.074, 0.089, and 0.119. We can see that, the network in **(a)** has a higher WC than that of **(b)**, while also displaying a larger GE. The corresponding relation between WC and GE becomes more prominent for the third phase $10^2 \sim 10^3$, when the training accuracy of the network has reached a certain level.

Testing our hypothesis on a range of networks has confirmed it: we have observed the same connection across different architectures, and we have observed it for both, convolutional neural networks (CNNs) and fully-connected networks (FCNs).

Based on this observation, this paper makes the following three major contributions. The first is to study if, and how, the PAC Bayesian framework (McAllester, 1999) on **the generalisation error bound can be upgraded** to incorporate WC. We observe that the current framework requires Kullback-Leibler (KL) divergence between the prior (before training) and the posterior (after training) distributions over $\Theta$, a high-dimensional multivariate random variable that represents the network parameters, with each dimension corresponding to a neuron. However, it is notoriously difficult to precisely estimate the KL quantity for high-dimensional multivariate variables (Singh and Póczos, 2017; Goldfeld et al., 2019), while evaluating KL with an off-the-shelf dimension-wise estimator such as Lombardi and Pant (2016); Kolchinsky and Tracey (2017) (by ignoring the cross-dimension correlation) can be arbitrarily inaccurate. To overcome these limitations, existing work (e.g. (Neyshabur et al., 2017; Jiang et al., 2020)) assumes that both the prior and the posterior distributions are Gaussian with each dimension being independent and identically distributed (i.i.d.). While it is reasonable to have an i.i.d. Gaussian distributed initialisation, it is unlikely that the components remain independent after training. To amend this, we show that the bound can be rectified under certain conditions by incorporating the WC. This theoretical result will enable us to tighter estimate the GE.

The second contribution is based on an observation over the new lifted PAC Bayesian bound. We show that the bound is monotonic with respect to the WC. More precisely, **our lifted PAC Bayesian bound decreases when the WC falls**. Moreover, this theoretical result aligns with our empirical observation from Figure 1. To fully understand—and exploit—the potential of this observation, we formalise a novel complexity measure by lifting the PAC Bayes measure (McAllester, 1999; Dziugaite and Roy, 2017) with the WC, and conduct experiments over a set of networks against a number of existing complexity measures. Our experimental results are very promising: the new measure provides the best ranking over the networks with respect to their generalisation error. That is, by comparing the values of the new measure, we are able to predict—in a more precise way than by using previous measures—which network has a better generalisation ability. Moreover, calculating the value of our new measure is cheap (quadratic in the representation of the weights). This makes it feasible to estimate the generalisation ability of neural networks without resorting to a testing dataset. Our experimental results show our advancement to the state-of-the-art as described in (Chatterji et al., 2020; Jiang et al., 2020). In particular, Jiang et al. (2020) concluded that sharpness-based measures, such as sharpness PAC-Bayesian bounds, perform better overall and seem to be promising candidates for further research. Our results advance this and show that weight correlation can be used to improve the PAC-Bayesian bounds further.

The third contribution is motivated by the above results, but exploits them in a different way: we explore the possibility to **enhance the training process with the WC**. We formalise a novel regularisation term and conduct experiments on a spectrum of convolutional networks. Our experimental

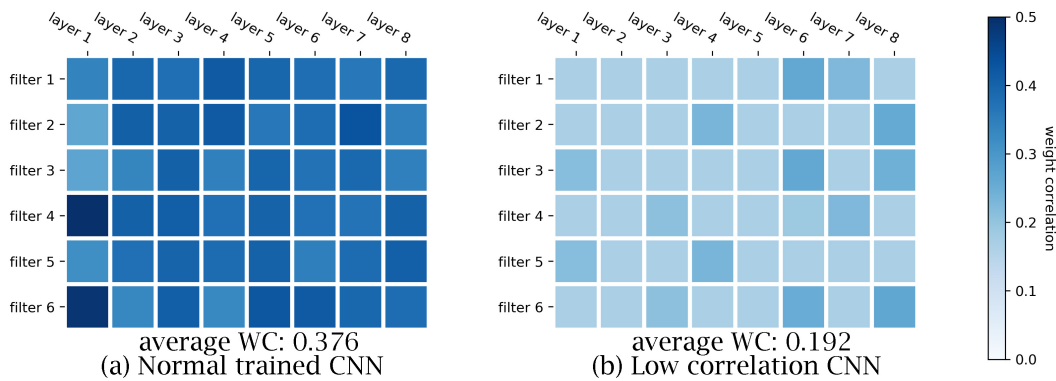

Figure 2: Visualisation of individual filters' weight correlations in two CNNs, whose structure is adapted from VGG16. The CNN in **(a)** is trained with the standard objective, while the one in **(b)** is trained using our new regularisation term (Section 5). Both CNNs are trained on the Fashion-MNIST dataset. The generalisation performance of **(b)** is better than **(a)** with a 0.8% improvement (87.1% vs 87.9%, which translates to a more than 6% higher error rate for **(a)**), which aligns with our expectations for the heatmaps: the left heatmap shows a significantly higher weight correlation for almost all filters compared to the right heatmap.

results show that the new regularisation term can **reduce the generalisation error** without compromising the training accuracy. This improvement is consistent across the models and datasets we have worked with.

## 2   Related Work

Evaluating the generalisation performance of neural networks has been a research focus since Baum and Haussler (1989). Many generalisation bounds and complexity measures have been proposed so far. Bartlett (1998) highlighted the significance of the norm of the weights in predicting the generalisation error. Since then, various analysis techniques have been proposed. They have either been based on covering number and Rademacher complexity (Bartlett et al., 2017; Neyshabur et al., 2018, 2015), or they have used approaches similar to PAC-Bayes (Neyshabur et al., 2017; Arora et al., 2018; Nagarajan and Kolter, 2019a; Zhou et al., 2018). A number of recent theoretical works have shown that, for a large network initialised in this way, accurate models can be found by traveling a short distance in parameter space (Du et al., 2019; Allen-Zhu et al., 2019). Thus, the required distance from the initialisation may be expected to be significantly smaller than the magnitude of the weights. Furthermore, there is theoretical reason to expect that, as the number of parameters increases, the distance from the initialisation falls. This has motivated works that focus on the role of the distance to initialisation rather than on the norm of the weights in generalisation (Dziugaite and Roy, 2017; Nagarajan and Kolter, 2019b; Long and Sedghi, 2020). Recently, Chatterji et al. (2020) introduced module criticality and analysed how different modules in the network interact with each other and influence the generalisation performance as a whole.

## 3   Definition of Weight Correlation

Many aspects of networks' design and analysis remain open challenges. One of the most important research questions is "what makes an architecture generalise better than others given a specific task?" While this paper does not aim to provide a comprehensive answer to this question, we argue that WC is a key factor that should be taken into account when designing and analysing networks. To this end, we define WC in this section, and suggest that WC should be considered in the PAC Bayesian framework (a theoretical framework to analyse the generalisability of machine learning models) in Section 4 and that WC should be considered during training in Section 5.

Let $\Theta = (\theta_1, ..., \theta_L)$ be the parameters of a network with $L$ layers, where $\theta_\ell$ refers to the parameters, including weight matrix $w_\ell$ and bias $b_\ell$, at layer $\ell$. To emphasise different architecture types, we may use the dedicated symbols $\mathbbm{w}$ (to represent a filter matrix in CNNs) and $\omega$ (to represent a weight matrix in FCNs). $\theta_\ell^0$ and $\theta_\ell^F$ refer to the value of parameters at initialisation and at the end of training,

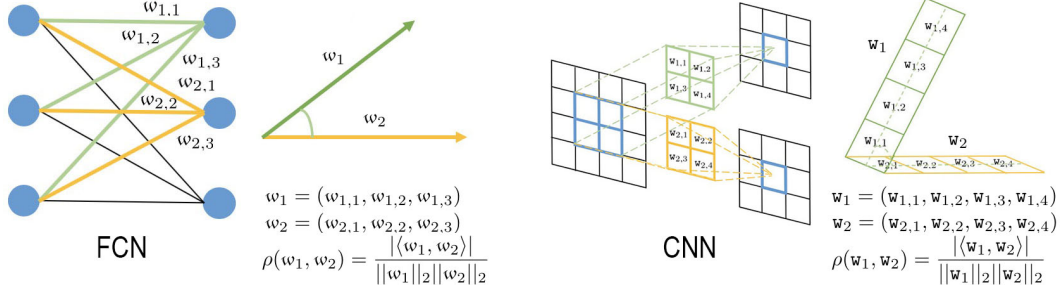

Figure 3: **(FCN)** The WC of any two neurons is the cosine similarity of the associated weight vectors. **(CNN)** The WC of any two filters is the cosine similarity of the reshaped filter matrices.

respectively, at layer $\ell$. Moreover, we write $N_\ell$ for the number of modules (either neurons or filters, depending on the network) at layer $\ell$. WC for different modules in FCN and CNN follows Def. 3.1 and Def. 3.2, respectively.

**Definition 3.1 (Weight Correlation in FCN)** *Given weight matrix $\mathrm{w}_\ell \in \mathbb{R}^{N_{\ell-1} \times N_\ell}$ of the $\ell$-th layer, the average weight correlation is defined as*

$$\rho(\mathrm{w}_\ell) = \frac{1}{N_\ell(N_\ell - 1)} \sum_{\substack{i,j=1 \\ i \neq j}}^{N_\ell} \frac{|\mathrm{w}_{\ell i}^T \mathrm{w}_{\ell j}|}{||\mathrm{w}_{\ell i}||_2 ||\mathrm{w}_{\ell j}||_2}, \tag{1}$$

*where $\mathrm{w}_{\ell i}$ and $\mathrm{w}_{\ell j}$ are $i$-th and $j$-th column of the matrix $\mathrm{w}_\ell$, corresponding to the $i$-th and $j$-th neuron at $\ell$-th layer, respectively. Intuitively, $\rho(\mathrm{w}_\ell)$ is the average cosine similarity between weight vectors of any two neurons at the $\ell$-th layer.*

**Definition 3.2 (Weight Correlation in CNN)** *Given the filter tensor $\mathrm{w}_\ell \in \mathbb{R}^{f \times f \times N_{\ell-1} \times N_\ell}$ of the $\ell$-th layer, where $f \times f$ is the size of the convolution kernel, $\mathrm{w}_{\ell i} \in \mathbb{R}^{f \times f \times N_{\ell-1}}$ and $\mathrm{w}_{\ell j} \in \mathbb{R}^{f \times f \times N_{\ell-1}}$ are the $i$-th and $j$-th filter, respectively, of the filter tensor $\mathrm{w}_\ell$. By reshaping $\mathrm{w}_{\ell i}$ and $\mathrm{w}_{\ell j}$ into $\mathrm{w}'_{\ell i} \in \mathbb{R}^{f^2 \times N_{\ell-1}}$ and $\mathrm{w}'_{\ell j} \in \mathbb{R}^{f^2 \times N_{\ell-1}}$, respectively, the weight correlation is defined as*

$$\rho(\mathrm{w}_\ell) = \frac{1}{N_\ell(N_\ell - 1)N_{l-1}} \sum_{\substack{i,j=1 \\ i \neq j}}^{N_\ell} \sum_{z=1}^{N_{l-1}} \frac{|\mathrm{w}'^T_{\ell i,z} \mathrm{w}'_{\ell j,z}|}{||\mathrm{w}'_{\ell i,z}||_2 ||\mathrm{w}'_{\ell j,z}||_2}, \tag{2}$$

*where $\mathrm{w}'_{\ell i,z}$ and $\mathrm{w}'_{\ell j,z}$ are the $z$-th column of $\mathrm{w}'_{\ell i}$ and $\mathrm{w}'_{\ell j}$ respectively. Intuitively, $\rho(\mathrm{w}_\ell)$ is defined as the cosine similarity between filter matrices.*

Although there are other correlation metrics for matrices, such as chordal distance and subspace colinearity (Yi and Au, 2011), we adopt the cosine similarity metric because of its low computational complexity—in Section 5 the WC will be computed in each epoch of training.

As an example, Figure 2 visualises the individual weight of filters' weight correlations. The above definition of $\rho(\mathrm{w}_\ell)$ simply computes the average over them. In turn, each individual filter's weight correlation is computed by averaging over its cosine similarities with other filters. Figure 3 illustrates the definitions with two simple examples.

## 4 Generalisation Bounds Incorporating Weight Correlation

We analyse a theoretical connection between WC and GE by deriving a generalisation bound using the PAC-Bayesian framework (McAllester, 1999). Given a prior distribution over the parameters $\Theta$, which is selected before seeing a training dataset, a posterior distribution on $\Theta$ will depend on both, the training dataset and a specific learning algorithm. The PAC-Bayesian framework bounds the generalisation error with respect to the Kullback-Leibler (KL) divergence (Kullback and Leibler, 1951) between the posterior and the prior distributions.

**Theorem 4.1** *(McAllester, 1999; Dziugaite and Roy, 2017) Consider a training data set $S$ with $m \in \mathbb{N}$ samples drawn from a distribution $D$. Given a learning algorithm (e.g., a classifier) $f_\Theta$ with*

*prior and posterior distributions $P$ and $Q$ on the parameters $\Theta$ respectively, for any $\delta > 0$, with probability $1 - \delta$ over the draw of training data, we have that*

$$\mathbb{E}_{\Theta \sim Q}[\mathcal{L}_D(f_\Theta)] \leq \mathbb{E}_{\Theta \sim Q}[\mathcal{L}_S(f_\Theta)] + \sqrt{\frac{\mathrm{KL}(Q||P) + \log \frac{m}{\delta}}{2(m-1)}}, \tag{3}$$

*where $\mathbb{E}_{\Theta \sim Q}[\mathcal{L}_D(f_\Theta)]$ is the expected loss on $D$, $\mathbb{E}_{\Theta \sim Q}[\mathcal{L}_S(f_\Theta)]$ is the empirical loss on $S$, and their difference yields the generalisation error.*

Theorem 4.1 outlines the role KL divergence plays in the upper bound of the generalisation error. In particular, a smaller KL term will help tighten the generalisation error bound. This is also our motivation for adding a regularisation term to improve generalisation performance in Section 5.

Below, we show how to incorporate WC into this framework. Assume that $P$ and $Q$ are Gaussian distributions with $P = \mathcal{N}(\mu_P, \Sigma_P)$ and $Q = \mathcal{N}(\mu_Q, \Sigma_Q)$, then the KL-term can be written as follows (details are given in Appendix A):

$$
\begin{aligned}
&\mathrm{KL}(\mathcal{N}(\mu_Q, \Sigma_Q)||\mathcal{N}(\mu_P, \Sigma_P)) \\
&= \frac{1}{2}\Big[\mathrm{tr}(\Sigma_P^{-1}\Sigma_Q) + (\mu_Q - \mu_P)^\top \Sigma_P^{-1}(\mu_Q - \mu_P) - k + \ln \frac{\det\Sigma_P}{\det\Sigma_Q}\Big],
\end{aligned} \tag{4}
$$

where $k$ is the number of parameters in $\Theta$.

Similar to (Neyshabur et al., 2017; Jiang et al., 2020), we assume that the parameters in the prior distribution $P$ are independent and identically distributed (i.i.d.). That is, we have $P = \mathcal{N}(\theta^0, \sigma^2 I)$. We point out that, after some epochs of training, it is unlikely the parameters remain i.i.d., although they may still be Gaussian distributed. As such, for posterior distribution $Q$, we take weight correlation into account. To this end, we consider the posterior distribution with the following covariance matrix.

**Definition 4.2 (Posterior Covariance Matrix $\Sigma_Q$)** *Given $\rho(w_\ell)$ in Def. 3.1 and Def. 3.2, we introduce a correlation matrix $\Sigma_{\rho(w_l)} \in \mathbb{R}^{N_\ell \times N_\ell}$, with diagonal elements being 1, and off-diagonal ones all $\rho(w_\ell)$. Then, the posterior corvariance matrix can be represented as $\Sigma_{Q_{w_\ell}} = \Sigma_{\rho(w_\ell)} \otimes \sigma_\ell^2 I_{N_{\ell-1}}$ for both FNN and CNN models (See Appendix B for details), where $\otimes$ is Kronecker product.*

By Def. 4.2, we define the weight posterior distribution as $Q_w = \mathcal{N}(w^F, \Sigma_{\rho(w)} \otimes \sigma^2 I)$ and the bias posterior distribution as $Q_b = \mathcal{N}(b^F, \sigma^2 I)$. We remark that the assumption on Gaussian posterior distribution relaxes the i.i.d. assumption made in prior works (Dziugaite and Roy, 2017; Neyshabur et al., 2017; Jiang et al., 2020). This puts us in a sweet spot between the techniques that make unrealistic assumptions about the posterior distribution (usually i.i.d.), and approaches that make no assumptions, but only allow for an *a posteriori* estimation. (Moreover, such estimations are hard to compute and the existing methods are usually inaccurate for high dimensional data.) Then, the KL term can be simplified as a function of $\rho(w_\ell)$.

**Lemma 4.3** *Let $\mathbf{g}(w) = \sum \mathbf{g}(w_\ell)$ where $\mathbf{g}(w_\ell)$ defined by:*

$$\mathbf{g}(w_\ell) = -(N_\ell - 1)N_{\ell-1}\ln(1 - \rho(w_\ell)) - N_{\ell-1}\ln(1 + (N_\ell - 1)\rho(w_\ell)). \tag{5}$$

*Given the posterior covariance matrix in Def. 4.2, the KL term w.r.t. the $\ell$-th layer can be given by*

$$\mathrm{KL}(Q||P)_\ell = \frac{||\theta_\ell^F - \theta_\ell^0||_{\mathrm{Fr}}^2}{2\sigma_\ell^2} + \ln \frac{\det(\Sigma_{P_w})_\ell \cdot \det(\Sigma_{P_b})_\ell}{\det(\Sigma_{Q_w})_\ell \cdot \det(\Sigma_{Q_b})_\ell} = \frac{||\theta_\ell^F - \theta_\ell^0||_{\mathrm{Fr}}^2}{2\sigma_\ell^2} + \mathbf{g}(w_\ell). \tag{6}$$

*Further, when $\sigma_\ell^2 = \sigma^2$ for all $\ell$, we have $\mathrm{KL}(Q||P) = \sum_{\ell=1}^L \mathrm{KL}(Q||P)_\ell$.*

The proof of Lemma 4.3 is given in Appendix C. Given Lemma 4.3, we conclude that the KL term in (6) is positively correlated to the weight correlation $\rho(w_\ell)$.

**Corollary 4.4** *For a nontrivial network, i.e., $N_\ell > 1$ for all $\ell$, $\mathrm{KL}(Q||P)_\ell$ is positively correlated with $\rho(w_\ell) \in (0, 1)$, i.e., the decrease of $\rho(w_\ell)$ results in the decrease in $\mathrm{KL}(Q_w||P)_\ell$, since the gradient satisfies*

$$\frac{\partial \mathrm{KL}(Q||P)_\ell}{\partial \rho(w_\ell)} = \frac{N_{\ell-1}(N_\ell - 1)}{1 - \rho(w_\ell)} - \frac{N_{\ell-1}(N_\ell - 1)}{1 + (N_\ell - 1)\rho(w_\ell)} > 0. \tag{7}$$

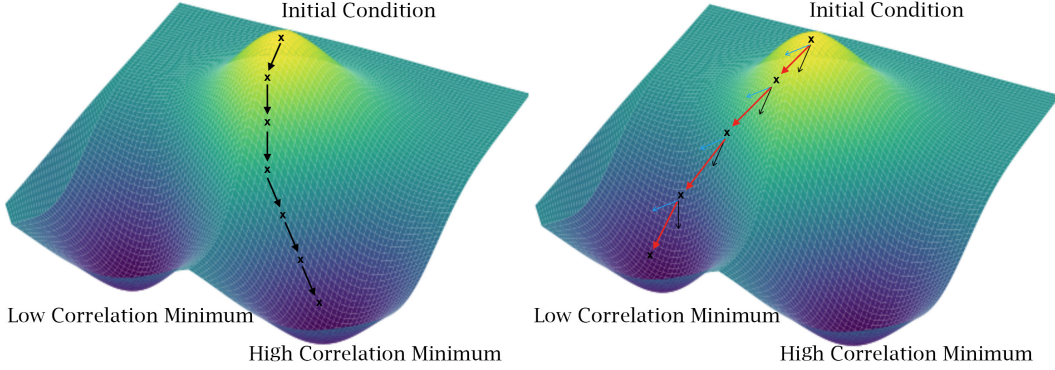

Figure 4: **Left**: Normal gradient-based optimisers may find a local minimum with high correlation. **Right**: Weight correlation regularisation helps the optimiser to find a low correlation minimum, which is more likely to be a global minimum.

The combination of Corollary 4.4 and Theorem 4.1 leads to our conclusion that reducing WC can tighten the PAC-Bayesian generalisation bound, and thus improve the generalisation performance, and vice versa. Notably, this conclusion aligns with the empirical observation we made in Figure 1, and will guide our design of regularisation in Section 5.

## 5 Regularisation Based on Weight Correlation

The previous sections have reached a clear view (supported by both theoretical results and empirical observations) that the WC is a key factor to the GE — our experimental results in Section 6.1 provide further evidence. In this section, we will explore how this new insight can be utilised to help the training process to learn a better model. To this end, we propose to use WC as a regulariser, similar to other regularisation techniques that have been widely available to the deep learning practitioner (Goodfellow et al., 2016; Srivastava et al., 2014).

The training of a neural network is seen as a process of optimising over an objective function $J(\theta; X, y)$, where $X$ is the input data, $y$ is the corresponding label, and $\theta$ is the parameter. Based on the positive correlation between WC and GE, we add a WC penalty term $\mathbf{g}(w) = \sum_\ell \mathbf{g}(w_\ell)$, which is a function of $\rho(w)$, to the objective function $J$, and denote the regularised objective function by $\tilde{J}$:

$$\tilde{J}(\theta; X, y) = J(\theta; X, y) + \alpha\mathbf{g}(w), \tag{8}$$

with the corresponding parameter gradient

$$\nabla_\theta \tilde{J}(\theta; X, y) = \nabla_\theta J(\theta; X, y) + \alpha\nabla_w \mathbf{g}(w), \tag{9}$$

where $\alpha \in [0, \infty)$ is a hyper-parameter that balances the relative contribution of the WC penalty term. For the weight matrix $w_\ell \in \mathbb{R}^{N_{\ell-1} \times N_\ell}$ from layer $\ell - 1$ to layer $\ell$,

$$\nabla_{w_\ell} \mathbf{g}(w_\ell) = \left[\frac{N_{\ell-1}(N_\ell - 1)}{1 - \rho(w_\ell)} - \frac{N_{\ell-1}(N_\ell - 1)}{1 + (N_\ell - 1)\rho(w_\ell)}\right]\frac{\partial\rho(w_\ell)}{\partial w_\ell}, \tag{10}$$

where $\frac{\partial\rho(w_\ell)}{\partial w_\ell}$ is a function of the $\ell_2$-norm of the vectors in $w_\ell$. (Details of $\frac{\partial\rho(w_\ell)}{\partial w_\ell}$ are given in Appendix D.)

When our training algorithm minimises the regularised objective function $\tilde{J}$, it will decrease both the original objective $J$ on the training data and the WC penalty term $\mathbf{g}(w)$. Different choices of the WC $\rho(w)$ can result in different solutions being preferred. In this paper, we use two kinds of $\rho(w)$ for FCNs and CNNs respectively, as defined in the Def. 3.1 and 3.2. Figure 4 provides an illustrative diagram to show the utility of the regularisation.

As an example, Figure 2 presents the visualisation of the resulting weight correlations of individual filters for two CNNs of the same structure. The two CNNs are trained without and with our regularisation term, respectively. We can see that the one trained with our regularisation term has significantly smaller correlations, and thus better generalisation ability.

Table 1: Complexity Measures (Measured Quantities)

| | |
|---|---|
| Generalisation Error (GE) | $\mathcal{L}_D(f_{\theta^F}) - \mathcal{L}_S(f_{\theta^F})$ |
| Product of Frobenius Norms (PFN) | $\prod_\ell \|\theta_\ell^F\|_{\text{Fr}}$ |
| Product of Spectral Norms (PSN) | $\prod_\ell \|\theta_\ell^F\|_2$ |
| Number of Parameters (NoP) | Total number of parameters in the network |
| Sum of Spectral Norms (SoSP) | Total number of parameters $\times \sum_\ell \|\theta_\ell^0 - \theta_\ell^F\|_2$ |
| Weight Correlation (WC) | $\frac{1}{\ell} \sum_\ell \rho(w_\ell)$ |
| PAC Bayes (PB) | $\sum_\ell \|\theta_\ell^0 - \theta_\ell^F\|_{\text{Fr}}^2 / 2\sigma_\ell^2$ |
| PAC Bayes & Correlation (PBC) | $\sum_\ell (\|\theta_\ell^0 - \theta_\ell^F\|_{\text{Fr}}^2 / 2\sigma_\ell^2 + \mathbf{g}(w_\ell))$ |

Table 2: Complexity measures for CIFAR-10

| Network | PFN | PSN | NoP | SoSP | PB | PBC | WC | GE |
|---|---|---|---|---|---|---|---|---|
| FCN1 | 8.1e7 | 1.4e4 | 3.7e7 | 1.6e9 | 1.1e4 | 1.14e5 | 0.297 | 2.056 |
| FCN2 | 3.3e7 | 8.5e3 | 4.2e7 | 1.61e9 | 8.8e3 | 1.24e5 | 0.296 | 2.354 |
| VGG11 | 8.5e10 | 1.4e5 | 9.7e6 | 2.4e8 | 2.0e3 | 3.41e4 | 0.273 | 0.929 |
| VGG16 | 5.1e15 | 1.3e7 | 1.5e7 | 5.2e8 | 2.6e3 | 3.73e4 | 0.275 | 0.553 |
| VGG19 | 1.1e19 | 2.9e8 | 2.1e7 | 8.1e8 | 3.3e3 | 4.26e4 | 0.274 | 0.678 |
| ResNet18 | 2.5e22 | 1.1e12 | 1.1e7 | 8.4e8 | 4.7e3 | 1.34e5 | 0.732 | 2.681 |
| ResNet34 | 9.9e34 | 4.9e16 | 2.1e7 | 3.1e9 | 1.0e4 | 1.30e5 | 0.733 | 2.552 |
| ResNet50 | 1.4e76 | 7.5e46 | 2.3e7 | 6.1e9 | 1.6e7 | 1.62e7 | 0.278 | 2.807 |
| DenseNet121 | 5.9e176 | 1.4e151 | 6.8e6 | 1.5e10 | 1.0e9 | 1.04e9 | 0.357 | 1.437 |
| Concordant Pairs | 21 | 21 | 22 | 26 | 24 | **29** | 24 | - |
| Discordant Pairs | 15 | 15 | 14 | 10 | 12 | **7** | 12 | - |
| Kendall's $\tau$ | 0.16 | 0.16 | 0.22 | 0.44 | 0.33 | **0.61** | 0.33 | - |

## 6 Experiments

We have conducted experiments to study the effectiveness of the new complexity measure in predicting GE (Section 6.1 and the effectiveness of exploiting WC during training to reduce GE (Section 6.2).

### 6.1 Complexity Measures

Following (Chatterji et al., 2020), we have trained several networks on the CIFAR-10 and CIFAR-100 datasets to compare our network complexity measure to earlier ones from the literature. The CIFAR-10 and CIFAR-100 datasets both consist of $3 \times 32 \times 32$ coloured pixel images of 50,000 training examples and 10,000 testing examples. The tasks are to classify the images into 10 classes and 100 classes, respectively. For all experiments, implementation and architecture details are provided in Appendix E.

Table 1 summarises the complexity measures that are calculated in this section. For the last two measures, we use a constant $\sigma_\ell$ instead of using sharpness-like methods (Keskar et al., 2016; Chatterji et al., 2020), in order to ensure the complexity measures only depend on the network architecture and parameters. The quantity PSN was proposed by (Bartlett et al., 2017) and SoSP was proposed by (Long and Sedghi, 2019).

In Table 2, we compare the generalisation performance of several DNN architectures trained on the CIFAR-10 dataset. In particular, we compare the rankings proposed by the weight correlation measure; PAC Bayes & correlation measure; and complexity measures from the literature with the empirical rankings obtained in the experiment. To this end, we compute Kendall's $\tau$ correlation coefficient (Kendall, 1938), which is defined as follows:

$$\tau = \frac{\text{(number of concordant pairs)} - \text{(number of discordant pairs)}}{\text{number of pairs}}, \qquad (11)$$

where the pair $(x_1, y_1), (x_2, y_2)$ are concordant if $x_1 > x_2, y_1 > y_2$ or $x_1 < x_2, y_1 < y_2$; in contrast, they are said to be discordant if $x_1 > x_2, y_1 < y_2$ or $x_1 < x_2, y_1 > y_2$. This coefficient lies in the range between -1 and 1, where a high coefficient reflects a more similar rank. We find that Kendall's $\tau$ coefficient is highest among all measures for our PBC measure (cf. Table 2) and

Table 3: Comparison of different models with and without WCD

| Network | Fashion-MNIST | | | | MNIST | | | |
|---|---|---|---|---|---|---|---|---|
| | Loss | Error % | WC | Train Loss | Loss | Error % | WC | Train Loss |
| FCN3 | 0.365± 0.005 | 12.7± 0.5% | 0.309 | 0.165± 0.005 | 0.163± 0.005 | 4.1± 0.5% | 0.292 | 0.003± 0.002 |
| FCN3    + WCD | **0.356±0.005** | **12.1±0.5%** | **0.248** | | **0.128±0.005** | **3.5±0.5%** | **0.237** | |
| VGG11* | 0.356±0.005 | 13.1±0.5% | 0.383 | 0.315±0.005 | 0.062±0.005 | 2.0±0.5% | 0.481 | 0.055±0.005 |
| VGG11* + WCD | **0.341±0.005** | **12.3±0.5%** | **0.232** | | **0.060±0.005** | **1.9±0.5%** | **0.286** | |
| VGG16* | 0.349±0.005 | 13.0±0.5% | 0.395 | 0.297±0.005 | **0.050±0.005** | **1.4±0.5%** | 0.502 | 0.035±0.005 |
| VGG16* + WCD | **0.335±0.005** | **12.4±0.5%** | **0.272** | | 0.053±0.005 | 1.6±0.5% | **0.313** | |
| VGG19* | 0.322±0.005 | 11.6±0.5% | 0.348 | 0.275±0.005 | 0.058±0.005 | **1.6±0.5%** | 0.387 | 0.03±0.005 |
| VGG19* + WCD | **0.321±0.005** | **11.4±0.5%** | **0.172** | | **0.057±0.005** | **1.6±0.5%** | **0.158** | |

| Network | CIFAR-10 | | | | SVHN | | | |
|---|---|---|---|---|---|---|---|---|
| | Loss | Error % | WC | Train Loss | Loss | Error % | WC | Train Loss |
| FCN3 | **1.43±0.02** | 50.7±0.5 % | 0.301 | 0.68±0.05 | **0.69±0.01** | **20.5±0.5 %** | 0.303 | 0.33±0.02 |
| FCN3    + WCD | 1.44±0.02 | **50.2±0.5 %** | **0.232** | | 0.72±0.01 | **20.5±0.5 %** | **0.257** | |
| VGG11* | 1.182±0.005 | 42.8±0.5 % | 0.444 | 1.141±0.005 | 0.904±0.005 | 29.0±0.5 % | 0.413 | 0.895±0.005 |
| VGG11* + WCD | **1.176±0.005** | **42.2±0.5 %** | **0.256** | | **0.893±0.005** | **28.7±0.5 %** | **0.251** | |
| VGG16* | 1.096±0.005 | 39.0±0.5 % | 0.429 | 1.01±0.01 | 0.637±0.005 | 20.3±0.5 % | 0.407 | 0.595±0.005 |
| VGG16* + WCD | **1.065±0.005** | **37.7±0.5 %** | **0.228** | | **0.616±0.005** | **19.6±0.5 %** | **0.231** | |
| VGG19* | 1.135±0.005 | 40.5±0.5 % | 0.382 | 1.025±0.005 | 0.625±0.005 | 19.5±0.5 % | 0.421 | 0.59±0.005 |
| VGG19* + WCD | **1.095±0.005** | **38.8±0.5 %** | **0.225** | | **0.612±0.005** | **19.1±0.5 %** | **0.221** | |

- the PB measure correctly ranks the generalisation performance of the networks FCN1, FCN2, VGG16, VGG19, and ResNet50;

- the WC measure correctly ranks the networks VGG16 and ResNet34; and

- combining the strengths of the above two measures, our PBC measure correctly ranks the networks FCN1, FCN2, VGG16, VGG19, ResNet34, and ResNet50.

We have also repeated the above experiment for the same networks trained on the CIFAR-100 dataset (see Appendix F). In this experiment, the PBC measure still provides a convincing performance.

## 6.2 Weight Correlation Descent Method (WCD)

We have trained weight correlation descent (WCD) neural networks for classification problems on datasets in different domains through the regularisation method from Section 5. We found that WCD improves the generalisation performance on some levels compared to neural networks that did not use this method. On account of high complexity, we have reduced the channel width of some networks used in the experiment. That is, compared with standard VGG, VGG* has decreased to 4, 4, 4, 8, 8 channels in the five stages. It turns out that the role of WC is not changed by such a channel reduction.

In this section, we present some key results that show the effectiveness of WCD. We have chosen Fashion-MNIST, MNIST, CIFAR-10 and SVHN datasets to demonstrate that WCD is a helpful technique for improving neural networks. The fashion-MNIST and MNIST datasets consist of $28 \times 28$ grayscale pixel article images of 60,000 training examples and 10,000 testing examples. The tasks are to classify the images into 10 digit classes. The SVHN dataset consists of $3 \times 32 \times 32$ coloured pixel images of 73,257 training examples and 26,032 testing examples. The task is also to classify the images into 10 digit classes. We train the models with and without WCD converge to same-level training loss, Table 3 compares the generalisation performance on the above datasets with ReLU activation function. Numbers of better value are **bold**. In our experiments, we consider early stopping (Yao et al., 2007) by reporting the best loss and corresponding error across final training epochs. A more detailed description of the experiments we have conducted is provided in Appendix G.

For the Fashion-MNIST dataset, all neural networks for the permutation invariant setting that do use WCD get an improvement in generalisation performance. VGG16* without WCD achieves an error of about 13.0% and a loss of about 0.349. With WCD the error and loss reduce to 12.4% and 0.335, respectively. Differnt to this, the errors are pretty high—as the networks are small—for the CIFAR-10 and SVHN datasets. Here, using WCD can still slightly improve the generalisation ability. Furthermore, in the respective VGG11*, VGG16*, VGG19*, using WCD has also slightly reduces the loss, error, and module correlation.

The experimental results for the MNIST dataset are ambiguous. This is principally because the narrow GE gap making it difficult to improve GE performance. For instance, with training loss converging to 0.035 (without using WCD), VGG16* achieves an error of about 1.4% and a loss of about 0.050. With WCD, the error increases to 1.6% and the loss increases to 0.053. However, with

training loss converging to 0.003, FCN3 achieves an error of about 4.1% and a loss of about 0.163 (without using WCD). With WCD, the error and loss reduce to 3.5% and 0.128. For some other networks, using WCD has also improved the generalisation performance.

## 7   Conclusion and Future Work

We have introduced weight correlation and discussed its importance to the generalisation ability of neural networks. In particular, we have injected WC into the popular PAC-Bayesian framework to derive a closed-form expression of the generalisation gap bound with mild assumption on weight distribution, and then employed it as an explicit regulariser—weight correlation decent (WCD)—to enhance generalisation performance within training. It turns out that WCD is an effective and computationally efficient tool to enhance generalisation performance in practice, which could complement other commonly used regularisers such as weight decay and dropout. More remarkably, considering weight correlation has proven to significantly enhance the complexity measure—that predicts the ranking of networks with respect to their generalisation errors—and the regularisation—that improves the generalisation performance of the trained model.

There are still some interesting directions to further explore the full strength of weight correlation for generalisation, robustness, and stability in both theory and practice. While Gaussian posterior distribution is assumed for the sake of tractability, it is crucial to further explore the true posterior distribution of parameters, although it is difficult to compute both empirically and theoretically. Sampling methods for the estimation of high-dimensional posterior distributions, such as Markov chain Monte Carlo (MCMC) sampling, could be employed here. Another practical issue concerns further complexity reduction of WCD: although WC is easy to compute in the forward propagation, computing WCD gradients in the back-propagation is more involved—which is crucial for the potential deployment of WCD in very large neural networks.

Moreover, given that the generalisation error is akin to model/software failure, it will be interesting to investigate whether techniques with stronger mathematical guarantees, such as formal verification based methods (Huang et al., 2017b), can be developed to improve over the popular PAC Bayesian bounds and, vice versa, whether complexity measures and regularisers can support the ultimate goal of "correct by construction" of deep learning systems.

## 8   Broader Impact

Our findings sharpen the theoretical and practical aspects of generalisation, one of the most important topics in machine learning: considering whether a trained model can be used on unseen data. Our findings can increase our understanding of the generalisation ability of deep learning and engender a broad discussion and in-depth research on how to improve the performance of deep learning.

This can unfold impact, first within the machine learning community and subsequently—through the impact of machine leaning—to other academic disciplines and the industrial sectors. Beyond the improvements that always come with better models, we also provide a better estimation of the generalisation error, which in turn leads to improved quality guarantees. This will enlarge the envelope of applications where deep neural networks can be used; not by much, maybe, but moving the goalposts of a vast field a little has a large effect.

A different kind of impact is that (neuronal) correlation is a concept, which is well studied in neuroscience. Our results could therefore lead to follow-up research that re-visits the connection between deep neural networks and neuroscience concepts.

**Acknowledgement**   GJ is supported by a University of Liverpool PhD scholarship. SS is supported by the UK EPSRC project [EP/P020909/1], and XH is supported by the UK EPSRC projects [EP/R026173/1,EP/T026995/1]. Both XH and SS are supported by the UK Dstl project [TCMv2]. 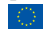 This project has received funding from the European Union's Horizon 2020 research and innovation programme under grant agreement No 956123. This project is supported by the Guangdong Science and Technology Department (Grant no. 2018B010107004), and NSFC (Grant Nos. 61761136011,61836005).

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
