[Supplementary Material]

## Appendix: Supplementary material

## A  Detailed Derivation of Formula 4

We state the PAC-Bayes theorem (Section 4) which bounds the generalization error of any posterior distribution $Q$ on parameters $\Theta$ that can be reached using the training set given a prior distribution $P$ on parameters that should be chosen in advance and before observing the training set. Let $Q$ and $P$ be $k$-dimensional Gaussian distributions (Jiang et al., 2020), the KL-term can be simply written as

$$
\begin{aligned}
\mathrm{KL}(\mathcal{N}(\mu_Q, \Sigma_Q) || \mathcal{N}(\mu_P, \Sigma_P)) &= \int [\ln(Q(x)) - \ln(P(x))]Q(x)dx \\
&= \int [\frac{1}{2}\ln\frac{\det\Sigma_P}{\det\Sigma_Q} - \frac{1}{2}(x-\mu_Q)^T\Sigma_Q^{-1}(x-\mu_Q) + \frac{1}{2}(x-\mu_P)^T\Sigma_P^{-1}(x-\mu_P)]Q(x)dx \\
&= \frac{1}{2}\ln\frac{\det\Sigma_P}{\det\Sigma_Q} - \frac{1}{2}\mathrm{tr}\{\mathbb{E}[(x-\mu_Q)(x-\mu_Q)^T]\Sigma_Q^{-1}\} + \frac{1}{2}\mathrm{tr}\{\mathbb{E}[(x-\mu_P)^T\Sigma_P^{-1}(x-\mu_P)]\} \\
&= \frac{1}{2}\ln\frac{\det\Sigma_P}{\det\Sigma_Q} - \frac{1}{2}\mathrm{tr}(I_k) + \frac{1}{2}(\mu_Q - \mu_P)^T\Sigma_P^{-1}(\mu_Q - \mu_P) + \frac{1}{2}\mathrm{tr}(\Sigma_P^{-1}\Sigma_Q) \\
&= \frac{1}{2}\left[\mathrm{tr}(\Sigma_P^{-1}\Sigma_Q) + (\mu_Q - \mu_P)^\top\Sigma_P^{-1}(\mu_Q - \mu_P) - k + \ln\frac{\det\Sigma_P}{\det\Sigma_Q}\right]
\end{aligned}
$$

where $\mathrm{tr}(\cdot)$ and $\det(\cdot)$ are the trace and the determinant of a matrix, respectively.

## B  Posterior Covariance Matrix

Given the weight matrix $w_\ell \in \mathbb{R}^{N_{\ell-1} \times N_\ell}$ with $i$-th column $w_{\ell i}$ as a random vector, the posterior covariance matrix $\Sigma_{Q_{w_\ell}}$ is defined in a standard way as $\Sigma_{Q_{w_\ell}} = \mathbb{E}[\mathrm{vec}(w_\ell)\mathrm{vec}(w_\ell)^T] \in \mathbb{R}^{N_\ell N_{\ell-1} \times N_\ell N_{\ell-1}}$, where $\mathrm{vec}(\cdot)$ is the vectorisation of a matrix. The $(i,j)$-th block is $[\Sigma_{Q_{w_\ell}}]_{i,j} = \mathbb{E}[w_{\ell i}w_{\ell j}^T] \in \mathbb{R}^{N_{\ell-1} \times N_{\ell-1}}$. For computational simplicity, we use the arithmetic mean instead of the expected value, so that the weight correlation $\rho(w_\ell)$ can be used to represent $[\Sigma_{Q_{w_\ell}}]_{i,j} = \rho(w_\ell)\sigma_\ell^2 I_{N_{\ell-1}}$.

Therefore we have $\Sigma_{Q_{w_\ell}} = \Sigma_{\rho(w_\ell)} \otimes \sigma_\ell^2 I_{N_{\ell-1}}$, where $\otimes$ is the Kronecker product, and the correlation matrix $\Sigma_{\rho(w_l)} \in \mathbb{R}^{N_\ell \times N_\ell}$ can be written as

$$
\Sigma_{\rho(w_\ell)} = \begin{bmatrix} 1 & \rho(w_\ell) & \rho(w_\ell) \\ \rho(w_\ell) & 1 & \rho(w_\ell) \\ \rho(w_\ell) & \rho(w_\ell) & 1 \end{bmatrix}_{N_\ell} .
$$

For simplicity, we use the same average $\rho(w_\ell)$ for layer $\ell$. Thus, $\Sigma_{Q_{\mathbf{w}_\ell}}$ is a $N_{\ell-1}N_\ell \times N_{\ell-1}N_\ell$ matrix with some elements being $\rho(w_\ell)$. For example, letting $N_{\ell-1}$ be 2 and $N_\ell$ be 3, we have

$$
\Sigma_{Q_{w_\ell}} = \Sigma_{\rho(w_\ell)} \otimes \sigma_\ell^2 I_2 = \begin{bmatrix}
\sigma_\ell^2 & 0 & \rho(w_\ell)\sigma_\ell^2 & 0 & \rho(w_\ell)\sigma_\ell^2 & 0 \\
0 & \sigma_\ell^2 & 0 & \rho(w_l)\sigma_\ell^2 & 0 & \rho(w_l)\sigma_\ell^2 \\
\rho(w_\ell)\sigma_\ell^2 & 0 & \sigma_\ell^2 & 0 & \rho(w_\ell)\sigma_\ell^2 & 0 \\
0 & \rho(w_\ell)\sigma_\ell^2 & 0 & \sigma_\ell^2 & 0 & \rho(w_\ell)\sigma_\ell^2 \\
\rho(w_\ell)\sigma_\ell^2 & 0 & \rho(w_\ell)\sigma_\ell^2 & 0 & \sigma_\ell^2 & 0 \\
0 & \rho(w_\ell)\sigma_\ell^2 & 0 & \rho(w_\ell)\sigma_\ell^2 & 0 & \sigma_\ell^2
\end{bmatrix} .
$$

As the computation of true posterior is a notoriously hard problem (due to high-dimensional data, highly nonlinear network, etc), the assumption of Gaussian distribution is commonly used in the

literature to make reasoning more tractable. Nevertheless, we have contributed theoretically to better capture the true posterior by (1) relaxing an i.i.d. assumption made in [Dziugaite and Roy (2017); Jiang et al. (2020)] and (2) taking WC into account. We recognize that our hypothetical covariance only considers the linear correlation between weights of neurons (filters). There is a gap between our hypothetical covariance and true covariance. But we also remark that, an estimation of the "true" posterior from data is also problematic, (e.g., use sharpness-like methods (Keskar et al., 2016) to get samplings parameters and estimate the covariance), may easily lead to further question on the accuracy of estimation and intractable derivation in theory. All in all, this is an open question and needs further research.

## C Proof of Lemma 4.3

We start by stating the PAC-Bayes generalisation bound and introduce WC into the posterior distribution $Q$ of the KL term $\text{KL}(Q||P)$. Then, we prove that $\text{KL}(Q||P)_\ell = \frac{||\theta_\ell^F - \theta_\ell^0||_{\text{Fr}}^2}{2\sigma_\ell^2} + \mathbf{g}(w_\ell)$, and $\text{KL}(Q||P) = \sum \text{KL}(Q||P)_\ell$ when $\sigma_1 = \sigma_2 = \cdots = \sigma_L = \sigma$.

**Proof C.1 (Proof of Eq. 6)** *Let $(\Sigma_P)_\ell$, $(\Sigma_P)_\ell$ be the covariance matrix for $(P)_\ell$ and $(Q)_\ell$ respectively, $\Sigma_{P_w}$, $\Sigma_{P_b}$ be the covariance matrix for weights' prior distribution and bias' prior distribution respectively. Thus we have*

$$\text{KL}(Q||P)_\ell = \frac{||\theta_\ell^F - \theta_\ell^0||_{\text{Fr}}^2}{2\sigma_\ell^2} + \ln \frac{\det(\Sigma_P)_\ell}{\det(\Sigma_Q)_\ell}$$

$$= \frac{||\theta_\ell^F - \theta_\ell^0||_{\text{Fr}}^2}{2\sigma_\ell^2} + \ln \frac{\det(\Sigma_{P_w})_\ell \cdot \det(\Sigma_{P_b})_\ell}{\det(\Sigma_{Q_w})_\ell \cdot \det(\Sigma_{Q_b})_\ell}$$

$$= \frac{||\theta_\ell^F - \theta_\ell^0||_{\text{Fr}}^2}{2\sigma_\ell^2} + \ln \frac{\det(\sigma_\ell^2 I_{N_{\ell-1} N_\ell})}{\det(\Sigma_{\rho(w)} \otimes \sigma_\ell^2 I_{N_{\ell-1}})}$$

$$= \frac{||\theta_\ell^F - \theta_\ell^0||_{\text{Fr}}^2}{2\sigma_\ell^2} + \ln \frac{\sigma_\ell^{2N_{\ell-1} N_\ell}}{\sigma_\ell^{2N_{\ell-1} N_\ell}(1 - \rho(w_\ell))^{(N_\ell - 1)N_{\ell-1}}(1 + (N_\ell - 1)\rho(w_\ell))^{N_{\ell-1}}}$$

$$= \frac{||\theta_\ell^F - \theta_\ell^0||_{\text{Fr}}^2}{2\sigma_\ell^2} - (N_\ell - 1)N_{\ell-1} \ln(1 - \rho(w_\ell)) - N_{\ell-1} \ln(1 + (N_\ell - 1)\rho(w_\ell))).$$

**Proof C.2** *Let $\sigma_1 = \sigma_2 = \cdots = \sigma_L = \sigma$. Then we have*

$$\text{KL}(Q||P) = \frac{||\theta^F - \theta^0||_{\text{Fr}}^2}{2\sigma^2} + \ln \frac{\det \Sigma_P}{\det \Sigma_Q}$$

$$= \sum_{\ell=1}^{L} \frac{||\theta_\ell^F - \theta_\ell^0||_{\text{Fr}}^2}{2\sigma^2} + \ln \prod_{\ell=1}^{L} \frac{\det(\Sigma_P)_\ell}{\det(\Sigma_Q)_\ell}$$

$$= \sum_{\ell=1}^{L} \left( \frac{||\theta_\ell^F - \theta_\ell^0||_{\text{Fr}}^2}{2\sigma^2} + \ln \frac{\det(\Sigma_P)_\ell}{\det(\Sigma_Q)_\ell} \right)$$

$$= \sum_{\ell=1}^{L} \text{KL}(Q||P)_\ell.$$

## D Details of Regularisation

In Section 5, we propose a parameter gradient that is given by

$$\nabla_\theta \tilde{J}(\theta; X, y) = \nabla_\theta J(\theta; X, y) + \alpha \nabla_w \mathbf{g}(w),$$

where for the weight matrix $w_\ell \in \mathbb{N}^{N_{\ell-1} \times N_\ell}$ from layer $\ell - 1$ to layer $\ell$, we have

$$\nabla_{w_\ell} \mathbf{g}(w_\ell) = \left[ \frac{N_{\ell-1}(N_\ell - 1)}{1 - \rho(w_\ell)} - \frac{N_{\ell-1}(N_\ell - 1)}{1 + (N_\ell - 1)\rho(w_\ell)} \right] \frac{\partial \rho(w_\ell)}{\partial w_\ell}.$$

And for the element $w_{\ell,(i,j)}$ of $i$-th row and $j$-th column in $w_\ell$, we have

$$\frac{\partial \rho(w_\ell)}{\partial w_{\ell,(i,j)}} = \frac{1}{N_\ell - 1} \sum_{q=1,q\neq j}^{N_\ell} \left\{ \text{sign}(w_{\ell,(,j)}^T w_{\ell,(,q)}) \left[ \frac{w_{\ell,(i,q)}}{||w_{\ell,(,j)}||_2 ||w_{\ell,(,q)}||_2} - \frac{w_{\ell,(i,j)} w_{\ell,(,j)}^T w_{\ell,(,q)}}{||w_{\ell,(,j)}||_2^3 ||w_{\ell,(,q)}||_2} \right] \right\},$$

where $w_{\ell,(,j)}$ and $w_{\ell,(,q)}$ are $j$-th and $q$-th column in $w_\ell$, respectively.

# E  Details of the experiments, and the results using the CIFAR10 dataset

For all our experiments in Section 6.1, we use the CIFAR10 and CIFAR100 datasets. To train our networks we used Stochastic Gradient Descent (SGD) with momentum 0.9 to minimise multi-class cross-entropy loss with 0.01 learning rate and 500 epochs. We mainly study four types of neural network architectures:[1]

- Fully Connected Networks (FCNs): FCN1 contains 5000, 2500, 2500 and 1250 hidden units respectively while FCN2 contains 10000, 1000, 1000 and 1000 hidden units respectively. Each of these hidden layers is followed by a batch normalization layer and a ReLU activation. The final output layer has an output dimension of 10 or 100 (i.e., number of classes).
- VGGs: Architectures by Simonyan and Zisserman (2015), that consist of multiple convolutional layers, followed by multiple fully connected layers and a final classifier layer (with output dimension 10 or 100). We study the VGG networks with 11 and 16 layers.
- DenseNets: Architectures by Huang et al. (2017a) that consist of multiple convolutional layers, followed by a final classifier layer (with output dimension 10 or 100). We study the DenseNet with 121 layers.
- ResNets: Architectures used are ResNets V1 (He et al., 2016). All convolutional layers (except downsampling convolutional layers) have kernel size $3 \times 3$ with stride 1. Downsampling convolutions have stride 2. All the ResNets have five stages (0-4) where each stage has multiple residual/downsampling blocks. These stages are followed by a max-pooling layer and a final linear layer. We study the ResNet 18, 34, and 50.

In the experiment, we set a suitable upper bound $\mathbf{g}(w_l) \leq 50000$ for $\rho(w_l) \leq 1$, to avoid that $\mathbf{g}(w_l) \to +\infty$ when $\rho(w_l) \to 1$. For the PBC measure, letting $\sigma_\ell^2 = 1/L$, we replace $\sum_\ell (L\|\theta_\ell^0 - \theta_\ell^F\|_{\mathrm{Fr}}^2/2 + \mathbf{g}(w_\ell))$ with $\sum_\ell (\|\theta_\ell^0 - \theta_\ell^F\|_{\mathrm{Fr}}^2/2 + \mathbf{g}(w_\ell)/L)$. Details are given in the source codes.

# F  Details of the Results Using the CIFAR100 Dataset

In Table 4, we compare the generalisation performance of several DNN architectures trained on the CIFAR-100 dataset. We find that the PBC measure still has a convincing performance and successfully ranks the networks FCN1, VGG16, VGG19 and ResNet18.

Table 4: Complexity measures for CIFAR-100

| Network | PFN | PSN | NoP | SoSP | PB | PBC | WC | GE |
|---|---|---|---|---|---|---|---|---|
| FCN1 | 2.2e8 | 2.0e4 | 3.7e7 | 8.7e8 | 5.0e2 | 1.1e5 | 0.275 | 4.251 |
| FCN2 | 1.0e8 | 1.4e4 | 4.2e7 | 9.8e8 | 4.9e2 | 1.2e5 | 0.278 | 3.754 |
| VGG11 | 1.4e12 | 3.5e6 | 9.8e6 | 4.2e8 | 5.21e3 | 3.8e4 | 0.280 | 2.549 |
| VGG16 | 6.6e16 | 4.4e8 | 1.5e7 | 6.9e8 | 3.8e3 | 3.9e4 | 0.279 | 1.387 |
| VGG19 | 2.0e20 | 1.5e10 | 2.0e7 | 1.0e9 | 5.22e3 | 4.5e4 | 0.277 | 1.726 |
| ResNet18 | 1.0e24 | 4.5e12 | 1.1e7 | 8.6e8 | 5.3e3 | 1.5e5 | 0.764 | 5.756 |
| ResNet34 | 1.5e39 | 9.1e18 | 2.1e7 | 3.1e9 | 1.0e4 | 1.6e5 | 0.781 | 5.660 |
| ResNet50 | 9.1e76 | 1.0e46 | 2.4e7 | 5.2e9 | 1.1e6 | 1.1e6 | 0.278 | 4.320 |
| DenseNet121 | 1.4e192 | 4.2e153 | 6.9e6 | 1.6e10 | 1.9e9 | 1.9e9 | 0.389 | 4.583 |
| Concordant Pairs | 23 | 23 | 16 | 23 | 24 | **28** | 26 | - |
| Discordant Pairs | 13 | 13 | 20 | 13 | 12 | **8** | 10 | - |
| Kendall's $\tau$ | 0.27 | 0.27 | -0.11 | 0.27 | 0.33 | **0.55** | 0.44 | - |

## G  Effectiveness of WCD (details on experimental data)

Table 5: The architectures of FCN3, VGG11*, VGG16*, VGG19*

| FCN3 | FC-52 | FC-48 | FC-44 | FC-40 | FC-36 | FC-32 | FC-28 | FC-24 | FC-20 | FC-16 | | | | |
|---|---|---|---|---|---|---|---|---|---|---|---|---|---|---|
| VGG11* | conv3-4 | poolmax | conv3-4 | poolmax | conv3-4<br>conv3-4 | poolmax | conv3-8<br>conv3-8 | poolmax | conv3-8<br>conv3-8 | poolmax | FC-12 | FC-12 | FC-10 | soft-max |
| VGG16* | conv3-4<br>conv3-4 | poolmax | conv3-4<br>conv3-4 | poolmax | conv3-4<br>conv3-4<br>conv3-4 | poolmax | conv3-8<br>conv3-8<br>conv3-8 | poolmax | conv3-8<br>conv3-8<br>conv3-8 | poolmax | FC-12 | FC-12 | FC-10 | soft-max |
| VGG19* | conv3-4<br>conv3-4 | poolmax | conv3-4<br>conv3-4 | poolmax | conv3-4<br>conv3-4<br>conv3-4<br>conv3-4 | poolmax | conv3-8<br>conv3-8<br>conv3-8<br>conv3-8 | poolmax | conv3-8<br>conv3-8<br>conv3-8<br>conv3-8 | poolmax | FC-12 | FC-12 | FC-10 | soft-max |

As Table 5 shows, we used the above four networks to train the Fashion-MNIST, MNIST, CIFAR-10, SVHN datasets with ReLU activation function, 0.01 learning rate, stochastic gradient descent (SGD), 100 (FCN) or 500 (CNN) epochs, with and without WCD method. To converge to a same-level training loss, some models may be trained several times. Details are given in online code. FCN3 contains 52, 48, 44, 40, 36, 32, 28, 24, 20 and 16 hidden units respectively while VGG* consists of multiple convolutional layers with 4, 4, 4, 8, 8 channels in the five stages.

## H  Effectiveness of WCD (additional experiments on CIFAR-100 and Caltech-256)

Table 6: Comparison of different models with and without WCD

| Network | CIFAR-100 | | | | Caltech-256 | | | |
|---|---|---|---|---|---|---|---|---|
| | Loss | Error % | WC | Train Loss | Loss | Error % | WC | Train Loss |
| VGG11* | 3.138±0.005 | 76.3±0.5% | 0.379 | 2.86±0.05 | 4.807±0.005 | 90.1±0.5% | 0.461 | 0.795±0.05 |
| VGG11* + WCD | **3.043±0.005** | **75.0±0.5%** | **0.221** | | **4.779±0.005** | **89.6±0.5%** | **0.289** | |
| VGG16* | 3.090±0.005 | **75.5±0.5%** | 0.383 | 2.853±0.005 | 4.950±0.005 | 91.1±0.5% | 0.487 | 1.55±0.05 |
| VGG16* + WCD | **3.082±0.005** | 76.0±0.5% | **0.261** | | **4.746±0.005** | **88.9±0.5%** | **0.304** | |
| VGG19* | 3.058±0.005 | 75.8±0.5% | 0.348 | 2.83±0.02 | 4.966±0.005 | 90.9±0.5% | 0.365 | 1.515±0.005 |
| VGG19* + WCD | **3.043±0.005** | **75.7±0.5%** | **0.218** | | **4.698±0.005** | **88.6±0.5%** | **0.203** | |

The errors are pretty high and the results are more random–as the networks are small and datasets are more complicated–WCD still improves a little generalisation performance in most cases.

## Footnotes

[1]Code available at https://github.com/Alexkael/NeurIPS2020_Weight_Correlation.