[Reviews · NeurIPS 2020]

Review 1

Summary and Contributions: The paper introduces a simple measure for the generalization error of deep networks hereafter called WC. The WC measure concerns the average correlation of pairs of neurons in the same layer. The correlation is measured using cosine similarity. First it is shown that under the assumptions of an isotropic Gaussian prior and Gaussian posterior on the model parameters with a certain construction of covariance matrix using WC, the measure tightens the PAC-Bayes bound on the generalization. Second, they show that WC finds a better ranking (through Kendall correlation) of different trained models compared to standard PAC-Bayes-based measure. The comparisons are done using different architectures and training datasets. Finally, as common in the literature, the measure is used as an additional regularization term leading into some improvements over standard networks.

Strengths: + the WC measure is based on a simple notion of cosine similarity between pairs of neurons and yet achieves a noticeably good ranking of the trained networks based on their generalization errors. The simplicity of the notion and the clear connection to PAC-Bayes framework improve the chances of additional works in similar directions. + the paper addresses the three important aspects of a generalization measure, i.e., theoretical relevance, prediction abilities, and regularization effects.

Weaknesses: - The correlation of the proposed WC coefficient and the theoretical PAC-Bayesian generalization bound is set through the assumptions of the specifically-constructed setup of this work, i.e. an isotropic Gaussian prior, and a Gaussian posterior with covariance matrix formed by the WC coefficient component. This may not be a weakness in general but renders the theoretical part less insightful and informative. For instance, one more interesting (and less obvious to the reviewer) question would be how well this construction captures the true posterior. - Important recent methods that introduce measures to predict generalization error are missing from the related works section and notably the tables. For reference, the following paper surveys and benchmarks some of those recent methods: [a] Jiang et al., “Fantastic Generalization Measures and Where to Find Them”, ICLR 2020 - Figure 1 indicates that the WC coefficient monotonically increases as the training progresses (even when the generalization error is decreasing). In that regard the evolution of the measure during training is neither empirically or theoretically discussed.

Correctness: It seems to the reviewer that the paper is correct in the method and claims.

Clarity: + the paper is clearly written in general. - why does definition 3.2 --the WC coefficient for convolutional layers-- treat each slice of a convolutional filter separately? Wouldn’t it make more sense to treat the whole filter tensor/neuron? - section 6.1 refers to CIFAR categories as digit classes.

Relation to Prior Work: Please refer to the weaknesses section.

Reproducibility: Yes

Additional Feedback: ---- after rebuttal comment ---- We discussed the paper with other reviewers and the AC. We all had concerns regarding the theoretical aspect of the work but at different levels. I still believe that the significance of the experimental results and the simplicity of the measure and regularization technique are enough for the acceptance of the paper and thus keep my original rating. *** I strongly suggest the authors to use the extra page to thoroughly discuss the significance and relevance of the theory especially expanding on its limitations.


Review 2

Summary and Contributions: This paper studies the relation between weight correlation and generalization in deep nets. The weight correlation is the correlation (ie cosine) between the weight vectors of two neurons in a given layer (averaged over all pairs of neurons in that layer and summed over layers). This quantity seems empirically correlated with generalization, and the authors show that when using it as a regularizer, the test error of a trained network decreases for several datasets. They also derive a PAC-Bayesian bound based on this quantity.

Strengths: Soundness: - theoretical grounding: the proofs seem to be correct to the extent I could verify. - empirical evaluation: the experiments seem reasonable and appear to support the claim that weight correlation is a regularizer that improves generalization on several datasets. Significance and novelty: As far as I can tell, weight correlation has not be studied before in relation with generalization and the proposed regularizer is simple enough that it can be implemented easily. It could provide an interesting alternative to other generic regularizers such as weight decay. Relevance:

Weaknesses: Theoretical support: As many paper which propose a new regularizer, the derivation of a bound using this regularizer is not really providing "theoretical support" for the idea, unlike what authors seem to often imply. Indeed, for any regularizer, one can cook up a PAC-Bayesian bound that would exhibit this regularizer (by just building a prior using this regularizer). So, the real validation is via the experiments showing that the effect of the regularizer is to improve generalization. Regarding the claimed correlation between the regularizer and the generalization gap, Figure 1 is not really conclusive as the two curves show a monotonically decreasing training error, a monotonically increasing generalization gap and a monotonically increasing weight correlation. From those curves, one could claim that the training error itself is a good predictor of the generalization gap. Of course, table 2 shows that compared to other terms, the proposed one (PBC) is better correlated with generalization, although on CIFAR-100 (table 4) the gap with other measures is much smaller (PFN and PSN perform equally well).

Correctness: No issue regarding the claims, method and methodology.

Clarity: The paper is easily readable.

Relation to Prior Work: Relation to prior work is clear.

Reproducibility: Yes

Additional Feedback: POST REBUTTAL After reading the authors response and after the discussion I am still somewhat uncomfortable with the theoretical justification but can be convinced by the empirical evidence, so I keep my score as is.


Review 3

Summary and Contributions: The paper explores the following question. Does correlation among the parameters of a model (neural network) improve its generalization ability? The paper looks at this problem from both a theoretical and an experimental perspective. They provide some experimental evidence which helps answer the question in the affirmative.

Strengths: The paper raises an interesting question which hasn't been looked at before but which has been observed by the neuroscience community. Namely, correlation amongst the weights of the neurons helps generalization performance of the network. The authors do a good job of bringing this phenomenon to the attention of ML/Deep learning community. One experiment shows that wc measure can correctly rank the performance of some architectures on cifar-10 and cifar-100.

Weaknesses: While the big idea of weight correlation being important is a novel contribution. But the following limitations has led to vote for the paper not being accepted in its current form. 1. The theoretical claim that WC correlation is improves generalization ability follows from the earlier works on [ Dziugaite and Roy, 2017]. Cor. 4.4 which is the main theoretical result of the paper is a restatement of Thm. 4.1 under certain conditions rather than a major theoretical contribution. 2. Not enough discussions/justifications are provided for the particular forms of the posterior distributions (Section 4). 3. The authors propose a new regularizer based on weight correlation. However, more extensive empirical evaluation is needed to ascertain its effectiveness. On some of datasets in Section 6.2, the improvement in performance seemed very modest. 4. More experiments on larger datasets. The results presented are on datasets which are small or modest by modern standards. It would be interesting to repeat these sets of experiments on larger datasets.

Correctness: The experiments are correct but they do not conclusively answer the bigger question of whether weight correlation helps generalization.

Clarity: The paper is well written for the most part. However, some parts like the tables in experiments, more details could be added to improve the presentation.

Relation to Prior Work: Yes.

Reproducibility: Yes

Additional Feedback: I think this paper addresses an interesting question and takes the first steps in answering that question. However, more extensive empirical and theoretical investigation is needed to support the hypothesis that weight correlation indeed helps performance.


Review 4

Summary and Contributions: This paper studies *weight correlation*, a new generalisation measure. It is shown that the measure correlates well with generalisation error (on CIFAR-10 and -100) and that it can fruitfully be used as a regulariser (on MNIST, Fashion-MNIST, CIFAR-10 and SVHN).

Strengths: * Nicely done paper. Extremely clear writing and presentation of results. * Promising results (Table 2 and 3). * Code provided.

Weaknesses: * I worry that the claims about the measure being theoretically grounded are wrong, or at least misleading. The way I understand it, the paper introduces a method - WCD - which minimises weight correlation along with the loss. In order to provide performance guarantees like in Eq. (3) for this method, one would have to compute the posterior Q that WCD actually gives rise to. Instead, the paper defines a separate posterior, which is inspired by similar concepts, but essentially comes from nowhere and has no reason to be tied to WCD. I therefore find the discussion in Section 4 misleading. * Somewhat limited scope of experiments. E.g. do the findings hold up on ImageNet? * Limited comparison to prior work. For example, how does WC relate to some of the existing measures? How do the results in Table 2 relate to the findings in the Fantastic Measures [Jiang et al, ICLR'20] paper for example? * The method seems not to scale to large models, e.g. in the experiments of Table 3, only results for 'downsized' networks are presented. Minor: * Figure 1 is supposed to show how WC correlates with generalisation error. However, based on just these two plots, the difference in generalisation might just as well be due to the depth, or the width, or something else. All I'm saying is, Figure 1 is not much evidence for the claim. * In Lemma 4.3, it's not clear to me why there's sigma_l rather than sigma in the denominator. (I thought this should be the inverse covariance of the prior?) Also, doesn't the equality in line 152 (KL = sum of KLs) hold irrespective of sigma and sigma_l? My intuition is that KL is additive in independent factors, and the layers are assumed independent if I understand correctly. * In the experimental section [line 234], it says that the "best error and loss across all training epochs" is reported. This seems like cheating to me (one might end up inadvertently optimising on the test set). A safer way to do it would be to stop when the validation error is lowest, but then evaluate on a separate test set. * [Line 127] "distrituions"

Correctness: Claims regarding theory seem incorrect to me (mathematically correct, but incorrect interpretation). Claims regarding experiments seem justified.

Clarity: Yes, very well written.

Relation to Prior Work: Though I'm not worried about novelty per se, I'd appreciate more discussion of prior work (see Weaknesses above).

Reproducibility: Yes

Additional Feedback: I think the results are promising; it would be interesting to see if the findings hold up in larger scale experiments. ================ post-rebuttal ================== I thank the authors for their response. I appreciate the extra experiments on Caltech-256 and I'm overall enthusiastic about the experimental results. On the other hand, I wish the authors did not insist so strongly on the theory. After discussing with the other reviewers, I still think it's weak. PAC-Bayes is known as a bound-generating tool and can justify anything. Moreover, I worry that Thm 4.1 has little relevance for the proposed method, since it cannot be guaranteed that the actual posterior has the kind of skewed-covariance structure defined in Defn 4.2. Regarding computational efficiency, it is claimed in the rebuttal, but I see no evidence for it. All in all, I am on the fence. I see the paper's merits, but also think it could be improved.

[Author Response · NeurIPS 2020]

We are very grateful that all four reviewers recognise the novelty and originality of the paper. Below, we first clarify the
main contribution (in particular regarding Lemma 4.3), and then address the detailed points raised by the reviewers.

**Main Contribution** is the novel concept weight correlation (WC), which we believe is a key factor affecting the
generalisation ability. To consolidate this, we inject WC into the well-received PAC-Bayesian framework to derive a
*closed-form expression* of generalization gap bound with *mild assumption* on weight distribution, and then employ WC
as an explicit reguraliser to enhance generalisation performance within training. More importantly, the regulariser is
effective and computationally efficient in enhancing generalisation performance in practice.

Lemma 4.3 shows the positive correlation between WC and the bound (but not the generalisation itself). The effective-
ness of WC in either predicting the generalisation (Section 6.1) or training (Section 6.2) is shown with experiments.

The central question raised in the reviews is the justification of the assumption that we have a **Gaussian posterior**
distribution. We believe that this is a very mild assumption, which is partially justified by distributions tending to
converge against Gaussian distributions. More importantly, our assumption significantly relaxes the assumptions used
in prior works: an i.i.d. assumption made in [Dziugaite and Roy, 2017, Neyshabur et al. (2017); Jiang et al. (2020)].
Different to assuming a general Gaussian distribution, i.i.d. is unrealistic, such that we can lift an unrealistic assumption.

This also addresses the **R3: novelty of theoretical claim on WC correlation** concern: we have lifted the—unrealistic—
i.i.d. assumption from [Dziugaite and Roy, 2017], landing practical relevance to their findings.

This puts us in a sweet spot between the techniques that make unrealistic assumptions about the posterior distribution
(usually i.i.d.), and approaches that make no assumptions, but only allow for an a posteriori estimation. (Moreover, such
estimations are hard to compute and inaccurate for high dimensional data.) As a result, we have gained the capacity to
develop a regulariser, which is both meaningful and easy to compute.

**R1, R5: comparison with the state-of-the-art** While mainly focused on [Chatterji et al., ICLR'20], our results also
shed light on the fantastic measures paper [Jiang et al, ICLR'20]. In particular, in [Jiang, et al, ICLR'20], it is concluded
that "Sharpness-based measures such as sharpness PAC-Bayesian bounds ... perform better overall and seem to be
promising candidates for further research". Our results advance this and show that the PAC-Bayesian bounds can be
further improved with the weight correlation. Therefore, we believe our paper has advanced the state-of-the-art.

**R1, R2, R5: Figure 1** Figure 1 (in Introduction) is an illustrative example to show the positive correlation between WC
and generalisability. It is *not* to suggest a general trend for WC (the evolution of WC can be fluctuating) or a conclusive
result for the positive correlation. The latter is obtained by the theoretical and empirical results in the following sections.

**R2, R5: theoretical support of Lemma 4.3 to regulariser** We agree PAC-Bayes can only provide partial theoretical
support to a regulariser. Lemma 4.3 shows that WC is positively correlated with the generalisation bound. This result
suggests that it *may* be effective to consider WC as a regulariser. Then, extensive experiments are conducted to validate.

**R3: discussions/justification on the particular form of the posterior distributions** We guess you are referring to
Def. 4.2. Given the weight matrix $w_\ell \in \mathbb{R}^{N_{\ell-1} \times N_\ell}$ with $i$-th column $w_{\ell i}$ as a random vector, the posterior covariance
matrix $\Sigma_{Q_{w_\ell}}$ is defined in a standard way as $\Sigma_{Q_{w_\ell}} = \mathbb{E}[\text{vec}(w_\ell)\text{vec}(w_\ell)^T] \in \mathbb{R}^{N_\ell N_{\ell-1} \times N_\ell N_{\ell-1}}$, where $\text{vec}(\cdot)$ is the
vectorisation of a matrix. The $(i, j)$-th block is $[\Sigma_{Q_{w_\ell}}]_{i,j} = \mathbb{E}[w_{\ell i} w_{\ell j}^T] \in \mathbb{R}^{N_{\ell-1} \times N_{\ell-1}}$. For computational simplicity,
we use the arithmetic mean instead of the expected value, so that the weight correlation $\rho(w_\ell)$ can be used to represent
$[\Sigma_{Q_{w_\ell}}]_{i,j} = \rho(w_\ell)\sigma_\ell^2 I_{N_{\ell-1}}$. Therefore we have $\Sigma_{Q_{w_\ell}} = \Sigma_{\rho(w_\ell)} \otimes \sigma_\ell^2 I_{N_{\ell-1}}$, where $\otimes$ is the Kronecker product.

**R3: modest improvement in Section 6.2** While the improvement might look modest in individual cases, it is persistent
across the experiments.

**R3, R5: more experiments** We conducted additional experiments on Caltech-256 dataset for this rebuttal. It is
unrealistic to consider ImageNet, since a single training may take days (or months). The results on the Caltech-256
dataset are also promising, and similar to MNIST and CIFAR10. For complexity measure (like Table 2), we have
Kendall's $\tau$ at 0.33, as opposed to others at 0.28, 0.28, 0.17, 0.22. For the comparison of models with and without WCD
(like Table 3), those models with WCD achieve about 1% improvement on top-5 error over models without WCD.

**R5: WCD and posterior** We can confirm that our WCD is obtained from the expression of the posterior of PAC Bayes
(Lemma 4.3), and therefore there is no discrepancy between posterior of Lemma 4.3 and WCD-based posterior.

**R5: technical details of Lemma 4.3** There may be some misunderstanding to our technical details. For example, the
prior is a diagonal matrix $\text{diag}(\sigma_\ell^2)$ and the PCA Bayesian is conducted by summarising the results across layers.

**R5: termination condition of training** We use the same termination condition across all our experiments, so it is fair
to all methods. The reason for us to consider the best loss across the last few training epoch is to make sure that our
results are not affected by random factor – we find that there may still be uncertainty in the last few epochs.

[Meta-Review · NeurIPS 2020]

Inspired by a PAC-Bayes risk bound for Deep Neural Nets (DNNs) with a Gaussian posterior having a covariance matrix determined by the correlation between the weight vectors within the same layer, the authors propose a weight correlation descent algorithm for regularizing DNNs. The extensive numerical experiments provide a clear evidence of the advantage of reducing the correlation between the weight vectors within the same layer. We think that this regularizer, easy to implement, can provide an alternative (or be complementary) to other currently-used regularizers such as weight decay and drop-out.